# The Barriers to Sustainable Nutrition for Sustainable Health among Zayed University Students in the UAE

**DOI:** 10.3390/nu14194175

**Published:** 2022-10-07

**Authors:** Sharifa AlBlooshi, Alia Khalid, Rafiq Hijazi

**Affiliations:** 1Department of Health Sciences, College of Natural and Health Sciences, Zayed University, Dubai P.O. Box 19282, United Arab Emirates; 2Department of Mathematics and Statistics, College of Natural and Health Sciences, Zayed University, Abu Dhabi P.O. Box 144534, United Arab Emirates

**Keywords:** sustainable health and nutrition, healthy diet, nutritional knowledge

## Abstract

Unhealthy and unsustainable nutrition is a major concern globally, including in the United Arab Emirates. Although many education programs have been conducted, diet-related illnesses remain prevalent. This study aims to identify the barriers between knowledge and practice of sustainable healthy diets to achieve long-term health, among students of Zayed University in the United Arab Emirates. An online questionnaire was sent to Zayed University (ZU) students over 18 years of age, using snowball sampling. The participants achieved a mean score of 5.6 out of 11 in nutritional knowledge, and a mean score of 3.4 out of 6 in healthy habits. The only barrier that significantly affected dietary habits was not knowing how to plan a diet (*p* = 0.025). Accordingly, having good nutritional knowledge was significantly related to improved dietary habits (*p* < 0.001). In addition, school curriculums (*p* = 0.004), doing one’s own research (*p* < 0.001), and social media (*p* < 0.001) were significantly related to improved nutritional knowledge. The most commonly reported motivators for a healthier diet were “to keep their body healthy for a long time” and “to maintain a healthy weight” (72.6% and 70.1%, respectively). Overall, among ZU students the most significant barrier to achieving sustainable healthy nutrition was lack of knowledge. Education methods such as school curriculums, individual research, and social media were effective. Topics such as how to plan a diet, how to keep the body healthy, and how to maintain a healthy weight are of general interest.

## 1. Introduction

Sustainable nutrition contributes to the United Nation’s Sustainable Development Goals for 2030 [1]. It refers to sustainability at all phases of the food supply chain, including input production, agricultural production, food processing, distribution, cooking meals, and waste disposal. It involves five aspects: health, environment, economy, society, and culture [2]. This research focuses on the health aspects of sustainable nutrition, which aims at providing and ensuring nutritionally sufficient, safe, and healthy food [1]. Multiple studies have reported that consuming a diet that is nutritionally sufficient, safe, and healthy is essential for human growth and development, and prevents chronic morbidity. Pushing for sustainable nutrition and spreading awareness of the issue can bring the population’s attention to the consequences of unsustainable dietary habits and food choices on health and the environment [1,3]. 

To achieve sustainable health and nutrition, investigation of the population’s practices and perceptions is necessary to develop interventions and education programs that cater to their needs [1]. University students were selected as the focus of this study, because young people tend to establish life-long habits at this period in their lives as they become more independent [4].

Diet-related illnesses are a substantial public health issue worldwide, including in the United Arab Emirates. Data show that non-communicable diseases such as cardiovascular disease (CVD), diabetes mellitus, cancer, and chronic respiratory disease account for 77% of mortalities in the UAE. This is mainly attributed to the population’s unhealthy diet and sedentary lifestyle. Similarly, obesity, hypertension, hyperglycemia, and dyslipidemia are all highly prevalent [5]. 

Young people in the UAE display many unhealthy food habits. For instance, they consume fast foods, carbonated and sugary beverages, and other foods high in fat and added sugar, while also consuming inadequate amounts of fruit and vegetables. In addition, they tend to have irregular eating patterns, which may include skipping breakfast. These factors all contribute to the development of diet-related illnesses [6,7]. 

When it comes to the UAE and other Gulf countries, rapid urbanization and significant changes in the region’s socioeconomic status have led to the spread of poor dietary habits and lifestyles. For example, high-fat and calorie-dense foods have been made much more accessible and are perceived to be cheaper and more convenient [8,9]. Furthermore, the population may not be giving enough consideration to the long-term consequences of poor dietary habits. For example, obese adults in the UAE seem to consider appearance more than health to be a significant motivator for weight loss [10]. 

Pertaining to females specifically, studies in Arab countries revealed that women commonly find it difficult to fit in healthy meals while balancing the responsibilities arising from demanding jobs, family commitments, and conventional gender roles [4].

Among university students, many stated that they tended to choose unhealthy food options because of convenience. They indicated their belief that cooking a healthy meal is time-consuming, requires more effort, and costs more than buying fast food or ready-to-eat food. Students also commented that unhealthy and processed food was easily accessible on their university campus, through vending machines or stores [11,12]. 

A common pattern among these groups is their perception that unhealthy food options are less time-consuming, more convenient, and more accessible than healthy options. 

Furthermore, researchers have observed increases in poor dietary habits and sedentary behavior as students transition to college. During this period, students gain more autonomy and freedom over health-related choices such as diet and levels of physical activity. Additionally, they typically experience increased stress due to lifestyle changes and increased academic pressure [13,14,15]. Studies show that those who are stressed are prone to experience increased cravings and difficulty in regulating their appetite. They also tend to use food as a source of comfort and lean towards palatable foods that are high in sugar and fat [16,17,18]. Bad health habits that are developed at this stage of life are likely to continue into adulthood. Intervening at this stage can aid in preventing adulthood obesity [13,14,15].

Many education-based intervention programs have focused on increasing awareness of the significance of healthy eating and physical activity. For example, the UAE government has integrated this information into school curriculums [6]. As a result, research suggests that there is adequate knowledge and awareness in the UAE regarding the importance of a healthy diet and lifestyle. However, this knowledge is not being put into practice, as evidenced by the persistent prevalence of diet-related illnesses, unhealthy dietary habits, and sedentary lifestyles. Therefore, intervention focused on converting this knowledge into action is necessary [8,10]. 

Evidence-based interventions advise a multidisciplinary approach by including other health factors alongside diet, such as physical activity and behavioral therapy. Group-based interventions have also been proven effective as well as cost-efficient [19]. 

In conclusion, the education programs that are ongoing in the UAE may not be sufficient to achieve sustainable health by combating non-communicable diseases. These illnesses remain highly prevalent, and people continue to have poor diets. An evidence-based intervention needs to be developed that focuses more on the specific barriers to practicing healthy dietary habits. Although many studies have been carried out on the prevalence of the issue and people’s knowledge about it, little is known about what Emirati adults think would help them make healthier food choices and what they perceive as barriers to this. Perhaps, the key to sustainable health and nutrition lies in filling this gap. 

This study aims to determine what causes the gap between the knowledge and practice of sustainable healthy diets, among ZU students. 

## 2. Materials and Methods

### 2.1. Study Design

This study used a quantitative cross-sectional descriptive study design with a questionnaire as a data collection tool. This design was driven by two similar studies that also used questionnaires as measurement tools for cross-sectional descriptive research. The first of these examined university students’ knowledge and practice of sustainable nutrition [20], while the other looked at the UAE community’s perceptions of smoking, physical activity, and eating habits [8]. Both studies sought to investigate their population’s standpoint on specific health topics, in order to plan targeted interventions. 

### 2.2. Study Setting and Population

The current community-based study was conducted at Zayed University during the Spring 2022 semester. The study population was Zayed University students on both campuses (Dubai and Abu Dhabi). This population was chosen because young people tend to establish life-long habits at this period as they become more independent. We planned to recruit 1000 ZU students, and there was no dropout; students were recruited once only, and we finally enrolled a total of 1003 student participants. The PI sent the questionnaire via email to students and to other ZU faculties. It was shared with students and colleagues in different colleges at ZU. Moreover, faculties sent it to their students via email and WhatsApp groups. We used ZU email addresses to reach the participants. A total of 1096 questionnaires were completed by ZU students, questionnaires with missing answers were removed, and we collected 1003 completed questionnaires.

### 2.3. Sampling

The snowball sampling method was used, where the questionnaire link was disseminated by the authors via email and social media. The participants were asked to share the link with other students. The study was targeted at respondents from the Zayed University student population.

### 2.4. Data Collection and Measurements

A questionnaire based on the reviewed literature was administered to evaluate the accuracy of the population’s current perceptions, determine the barriers to sustainable health and nutrition, and investigate the degree of influence of educational programs on the population’s practices. The questionnaire was bilingual (English/Arabic) and consisted of 42 questions in four sections. The first section gathered sociodemographic information about the participants (14 questions). The second included questions about their nutritional knowledge (11 questions); these questions used validated, reliable General Nutrition Knowledge Questionnaires (GNKQs) as references [21,22]. The third section was about respondents’ current food habits (11 questions), using a similar style of questions to a previous study that also assessed food habits [8]. The final section asked for the study population’s perceptions and opinions regarding the improvement of their dietary habits (6 questions) [23,24,25,26].

The nutritional knowledge score was computed based on the 11 questions presented in Table 1. Each correct answer was assigned 1 point, generating a maximum score of 11 points. Nutritional habits were measured using 6 items, each assessed using six choices (3 times or more per day, 1–2 times per day, every two days, twice a week, once a week, never). A nutritional habit was considered healthy based on the frequency defined in Table 1. A participant was given one point for each healthy habit he or she indicated, resulting in a maximum possible score of 6 points.

### 2.5. Ethical Clearance

This study was conducted according to the guidelines of the Declaration of Zayed University and approved by the Research Ethics Committee (REC), Office of Research at Zayed University (protocol code: ZU22_024_F; date of approval: 14 March 2022). All participants joined the study voluntarily after signing an informed consent form. The form described the goals and objectives of the study and what was expected of the participants, and asked for their consent to the use and publishing of the data. Participants’ confidentiality and anonymity were maintained, and they were not subjected to any harm. 

### 2.6. Statistical Analysis

The collected data were analyzed using IBM SPSS Statistics 27 (IBM, Armonk, NY, USA). Descriptive statistics were employed to present categorical variables using frequencies and percentages, as well as continuous variables using mean and standard deviation.

An independent samples *t*-test and one-way ANOVA were carried out to compare mean scores for nutritional habits and nutritional knowledge across subgroups of sociodemographic variables. Normality of the scores for nutritional habits and knowledge was confirmed using the Shapiro–Wilk test, supporting the use of the aforementioned tests.

Four multiple linear regression models were constructed to identify the determinants of nutritional habits and knowledge among the participants. Two models were conducted using the full sample, while the other two models included participants who were not majoring in Public Health and Nutrition.

*p*-values less than 0.05 were considered statistically significant. 

## 3. Results

### 3.1. Demographic Information

Table 2 illustrates the demographic information of the participants. Most participants were between the ages of 18–22, representing 84.8% of the total. They were mostly female, at 95.5%, and UAE nationals, at 96.7%. The largest percentage resided in Dubai, at 69.3%, followed by 10.9% in Ajman and 9.2% in Abu Dhabi. Additionally, 84.0% lived in urban areas. In terms of socioeconomic status, the majority of participants were single, at 86.5%, and unemployed, at 81.5%. Consequently, 38.0% of the participants did not have a personal income or reported an income of less than 2000 AED (26.3%) or between 2000–5000 AED (20.8%).

In terms of education, 89.8% were pursuing an undergraduate degree. About one-third were from the College of Natural and Health Sciences (31.6%), followed by the College of Business (19.2%) and the College of Technological Innovation (15.2%). The most common majors were Public Health and Nutrition (14.1%), Environmental Science and Sustainability (11.0%), Information Technology (9.5%), and Psychology (9.1%). The combined percentage of business-related majors was 22.1%. Seventy-one percent of the participants reported that they had received education about healthy food and sustainable nutrition, most frequently from school curriculums, at 63.7%, and from social media, at 48.9%.

### 3.2. Nutritional Knowledge and Nutritional Habits

As shown in Table 3, the mean score of healthy nutritional habits was 3.42 (SD = 1.19). For nutritional knowledge, the mean score was 5.58 (SD = 2.43), which was approximately half of the total score. Additionally, most participants scored between 5 and 7 at 43.7%, followed by 35.2% who scored less than 4. Only 21.1% of the participants scored between 8 and 11.

The relationships between nutritional habits and knowledge and the socio-demographics of the participants are also presented in Table 3. Nutritional habits were significantly related to the emirate of residence (*p* = 0.046) and marital status (*p* = 0.017). Married participants and those from Umm Al Quwain scored higher than other groups. Meanwhile, nutritional knowledge was significantly related to gender (*p* < 0.001), degree level (*p* = 0.002), and employment status (*p* = 0.002). Overall, female, undergraduate, and unemployed participants showed higher levels of nutritional knowledge compared to other participants. 

### 3.3. Determinants of Nutritional Habits

The results of the multiple linear regression model for healthy habits are presented in Table 4. Firstly, the overall regression was statistically significant (*p* < 0.001, R-square = 7.24%). Secondly, the only statistically significant relationships were related to nutritional knowledge and education. Those who received education about healthy food and sustainable nutrition were more likely to have improved dietary habits (β = 0.233, *p* = 0.006). Additionally, those who had improved nutritional knowledge also had healthier diets (β = 0.075, *p* < 0.001). Among the perceived barriers, the only one that was significantly related to having less healthy habits was “I don’t know how to plan a diet” (β = −0.180, *p* = 0.025), which can also be considered a lack of knowledge.

To control for the effect of receiving formal nutritional knowledge, a new regression model was considered after excluding the participants majoring in public health and nutrition. Overall, the model was statistically significant (*p* < 0.001, R-square = 5.44%). However, only the informal nutritional education and knowledge were significant, while none of the listed barriers related to the healthy habits score. 

### 3.4. Effect of Variables on Nutritional Knowledge

To further investigate the determinants of nutritional knowledge, relationships with demographic characteristics and sources of knowledge were examined using a multiple linear regression model. The overall regression was statistically significant (*p* < 0.001, R-square = 14.4%). Table 5 shows that significant relationships were observed among demographic characteristics: being older in age (β = 0.072, *p* = 0.004), being female (β = 0.738, *p* = 0.040), being in undergraduate studies (β = −0.512, *p* = 0.045), and being unemployed (β = −0.433, *p* = 0.032). In terms of education, prior education about healthy food and sustainable nutrition was related to higher scores (β = 0.577, *p* = 0.005). The sources that were found to significantly increase this knowledge were school curriculums (β = 0.516, *p* = 0.004), one’s own research (β = 0.731, *p* < 0.001), and social media (β = 0.530, *p* < 0.001). Furthermore, being a Public Health and Nutrition major was also related to higher scores (β =1.422, *p* < 0.001). Nutrition students were possibly expected to score higher [27], so the variable was added to be controlled.

One of the common education-based interventions in the UAE is inclusion in school curriculums [6]. This study found that school curriculums are related to improved nutritional knowledge. Therefore, further studies should be carried out to examine whether the nutritional knowledge taught using current methods is retained, and whether it is put into practice.

In addition, social media and one’s own research were highly significant in relation to improved nutritional knowledge. This suggests that encouraging people to do their own research may target more effectively their individual concerns. Likewise, it seems that social media is effective in helping people improve their personal knowledge. Promoting media that has been created by trusted and valid sources and having it packaged in an easy to understand and enjoyable format may encourage people to learn more about sustainable health and nutrition.

Again, a new regression model was considered after excluding the participants majoring in public health and nutrition. The model was statistically significant (*p* < 0.001, R-square = 8.4%). The only noticeable change was the insignificance of the employment status (from *p* = 0.032 to *p* = 0.231). 

### 3.5. Perception, Opinions and Motivators Regarding Healthy and Sustainable Diets

To assess the attitudes of the population about solving the issue of unhealthy dietary practices, the participants answered questions regarding their perceptions of nutritional information, their motivators, and methods they believed would help them adopt healthier food habits. Table 6 illustrates the results. Firstly, 51.5% of the participants stated that healthy and sustainable nutrition is somewhat easily accessible to them, and 41.5% stated that it is very easily accessible. Only 7.0% said that it was not. Similarly, the majority (75.2%) stated that information about the topic is easy to find and understand. 

When asked about their motivators for eating a healthy diet, 72.6% stated that it was to keep their body healthy in the long term. At a close second, 70.1% said it was to maintain a healthy weight. Perhaps these are the areas that planners can target when creating interventions. 

Finally, in response to the question, “What do you think would help you and others eat healthier and more sustainable food?”, the two most frequent answers were “More variety” (52.7%) and “Ready-to-eat healthy meals” (52.5%). Another typical response was “Cheaper options” at 45.1%. 

## 4. Discussion

This study shows that there is no gap between the knowledge and practice of healthy habits; instead, knowledge and healthy dietary habits are both lacking. There was no significant relationship between commonly perceived barriers to healthy dietary habits, such as a busy rapid lifestyle, financial reasons, and inconvenience. Another observation is that having improved nutritional knowledge was significantly related to having improved dietary habits. 

The participants did not have adequate knowledge, evidenced by their mean score of 5.6 when asked to answer 11 GNKQ questions; while 43.7% of participants scored between 5 and 7, 35.2% scored less than 4, and only 21.1% scored between 8 and 11. This is despite the fact that 71.0% of the participants reported that they had received education about healthy dietary habits. This suggests that participants may not have retained the information. This is in contrast to other studies that found that UAE adults are knowledgeable about healthy dietary habits and nutrition [8,10]. The participants also had a mean of 3.4 healthy habits out of 6, consistent with other sources stating that the UAE population does not consume enough healthy foods [6].

The possible barriers that were stated in other studies, such as inconvenience, lack of time, and financial reasons [4,8,9,11], did not have a significant relationship with decreased healthy habits, although not knowing how to plan a diet did (β = −0.180, *p* = 0.025). This was against expectations, as studies have found that the UAE population chooses unhealthy options because they are more convenient, cheaper, and more accessible [8,9]. This may mean that the barriers do not affect the population as much as they perceive, or that there are other barriers that we do not know of that may be discovered using qualitative research with open-ended questions. Another possible explanation is that the issue for this sample was not accessibility, but rather variety, as 52.7% of the sample stated that more variety would help them consume healthier foods, while only 27.7% stated that greater availability would have this effect. Only 7.0% of the sample stated that they found healthy food options inaccessible. However, it was commonly reported that ready-to-eat healthy meals and cheaper options would help them eat more sustainably healthy diets (52.5% and 45.1%, respectively). This indicates that the inconvenience people may experience when preparing meals, and the cost of these foods, cannot be completely disregarded as barriers. As such, they can be targeted in efforts to achieve healthy and sustainable nutrition.

Nutritional knowledge was found to be a significant contributor to improved dietary habits (β = 0.075, *p* < 0.001), which is understandable as other studies have tied poor dietary habits to a lack of nutrition literacy [28]. To investigate further, the relations were examined of nutritional knowledge to a) demographic characteristics and b) sources of knowledge. Being older in age (β = 0.072, *p* = 0.004), being female (β = 0.738, *p* = 0.0.40), being in undergraduate studies (β = −0.512, *p* = 0.045), and being unemployed (β = −0.433, *p* = 0.032) were all associated with higher nutritional knowledge scores. In terms of sources of education, the fact that school curriculums were found to be significantly related to improved dietary habits (β = 0.516, *p* = 0.004) suggests that they are effective in increasing knowledge, but might not be for motivating healthy eating habits, as stated in other studies [6]. In addition, doing one’s own research (β = 0.731, *p* < 0.001) and social media (β = 0.530, *p* < 0.001) were highly significantly related to improved nutritional habits, showing that these methods can increase knowledge. This may be because they are usually initiated by the individual, meaning that they are interested and willing to learn. Another explanation could be that these methods target their individual concerns more effectively. Encouraging research and promoting media created by trusted and valid sources, packaged in an understandable and enjoyable format, may encourage people to learn more about sustainable health and nutrition.

The population’s motivations for maintaining a healthy diet were also assessed. The most commonly reported were “to keep their body healthy for a long time” and “to maintain a healthy weight” (72.6% and 70.1%, respectively). This result is in contrast with a study where obese adults in the UAE selected appearance as their most significant motivator [10]. In this population, 32.1% selected aesthetic reasons as a motivator. This may be due to differences in the two samples, as the sample in the previous study consisted of obese individuals only. Nevertheless, this shows that the UAE population is concerned with long-term health and understands the consequences of an unhealthy weight status, showing that there may be an interest in following more nutritionally sustainable diets. These points can be used as the focus to increase the effectiveness of interventions catered to this population. Interventions could also include instruction on how to plan a healthy diet. Those who stated that they did not know how to plan a diet had significantly less healthy habits. 

### Limitations

There were some potential drawbacks to this study. Firstly, the questionnaire consisted of close-ended questions, meaning that there might be other unexplored answers. Additionally, the choices might suggest answers that do not reflect the participants’ actual opinions. Additionally, because an online questionnaire was used, participants may have had an inaccurate understanding of some of the questions, or they might have interpreted them differently than intended. This limitation was minimized by using clear language and having the questions pilot-tested by students unfamiliar with the subject. 

Another limitation came from the sampling method. Snowball sampling follows a network of peers and may potentially lead to bias. Because the sample was not randomly selected, it was not possible to regulate sampling error. It cannot be guaranteed that the sample was representative of the population.

Another drawback is that there were insufficient data collected about the sample’s eating habits, such as serving size and food quality. Furthermore, information about other potential confounding variables, such as physical activity, weight, and health status, was lacking.

## 5. Conclusions

This study found that the most significant barrier to achieving sustainable healthy nutrition among Zayed University students was a lack of knowledge. Methods such as school curriculums, encouraging people to do their own research, and the use of social media were effective in increasing knowledge. Topics such as how to plan a diet, how to keep the body healthy, and how to maintain a healthy weight appear to be of interest and can be made focal points of education-based interventions to increase the likelihood of positive behavior practices. 

Addressing diet-related illnesses, by starting with dietary habits, is essential to achieving sustainable health and nutrition in the UAE. However, more research and targeted interventions are needed to combat this issue and prevent these diseases. This can lead to an improved utilization of healthcare resources and higher quality of life. Further studies can be undertaken to examine whether the nutritional knowledge taught using current methods is retained for a long period of time and if it is being utilized in practice.

## Figures and Tables

**Table 1 nutrients-14-04175-t001:** Point assignment of nutritional knowledge and healthy habits.

	Nutritional Knowledge Questions
1	Which of the following oils is the healthiest?
2	Which of the following is NOT a good source of protein?
3	Compared to fresh foods, highly processed foods are:
4	Compared to whole milk, skimmed milk has:
5	Foods labeled “light” or “diet” are always the healthier option.
6	Which of the following can prevent heart disease?
7	What can help reduce your risk of getting cancer?
8	Which of the following can help prevent diabetes?
9	A high protein diet will ensure you maintain a healthy weight.
10	Which of the following will raise your blood sugar levels the fastest?
11	The method used to prepare food does NOT affect its nutritional value.
	**Healthy Habits**
1	Eating fruits at least once to twice a day
2	Eating vegetables at least once to twice a day
3	Eating fish at least twice a week
4	Consuming at least as many whole grains as refined grains
5	Eating dairy products at least 1–2 times a day
6	Eating red meat no more than twice a week

**Table 2 nutrients-14-04175-t002:** Demographic information of the participants (n = 1003).

Characteristic	Categories	Count	Percentage %
Age	18–22	851	84.8%
	23–26	92	9.2%
	27–29	24	2.4%
	>30	36	3.6%
Gender	Male	45	4.5%
	Female	958	95.5%
Nationality	UAE National	969	96.7%
	Non UAE National	33	3.3%
Emirate of residence	Abu Dhabi	92	9.2%
	Dubai	695	69.3%
	Sharjah	51	5.1%
	Ajman	109	10.9%
	Umm Al Quwain	25	2.5%
	Ras Al Khaimah	19	1.9%
	Fujairah	12	1.2%
Residential area	Urban	843	84.0%
	Rural	160	16.0%
Marital status	Single	868	86.5%
	Married	124	12.4%
	Divorced	7	0.7%
	Separated	4	0.4%
Employment status	Employed (full-time/part-time/self-employed)	186	18.5%
	Unemployed/not working	817	81.5%
Personal monthly income (AED)	<2000	264	26.3%
	2000–5000	209	20.8%
	>5000	149	14.9%
	Not applicable	381	38.0%
Degree	Undergraduate	901	89.8%
	Postgraduate	102	10.2%
College	Arts and Creative Enterprises	89	8.9%
	Business	193	19.2%
	Communication and Media Sciences	92	9.2%
	Humanities and Social Sciences	96	9.6%
	Natural and Health Sciences	317	31.6%
	Education	64	6.4%
	Technological Innovation	152	15.2%
Major	Public Health and Nutrition	141	14.1%
	Psychology	91	9.1%
	Environmental Science and Sustainability	110	11.0%
	Information Technology	95	9.5%
	Human Resource Management	35	3.5%
	Marketing	37	3.7%
	Finance	46	4.6%
	Business Intelligence	81	8.1%
	Graphic Design	48	4.8%
	Education	51	5.1%
	International Relations	62	6.2%
	Communication and Media Sciences	34	3.4%
	Other	172	17.1%
Received education related to healthy food and sustainable nutrition	Yes	712	71.0%
	No	291	29.0%
Source of received education related to healthy food and sustainable nutrition	School curriculums	639	63.7%
	Nutritionist or dietitian	222	22.1%
	I did my own research	355	35.4%
	Social media	490	48.9%
	I have not learnt about it	132	13.2%

**Table 3 nutrients-14-04175-t003:** Relationship of nutritional knowledge and nutritional habits with socio-demographic factors (n = 1003).

		Nutritional Habits		Nutritional Knowledge	
Characteristics	Categories	Mean ± SD	*p*-Value	Mean ± SD	*p*-Value
Overall		3.42 ± 1.19		5.58 ± 2.43	
Age *	18–22	3.41 ± 1.19	0.406	5.58 ± 2.43	0.908
	23–26	3.51 ± 1.14		5.46 ± 2.55	
	27–29	3.17 ± 1.31		5.58 ± 2.28	
	>30	3.64 ± 1.29		5.81 ± 2.15	
Gender	Male	3.47 ± 1.12	0.780	4.38 ± 1.90	<0.001
	Female	3.42 ± 1.19		5.64 ± 2.44	
Nationality	UAE National	3.42 ± 1.19	0.810	5.57 ± 2.44	0.105
	Non UAE National	3.36 ± 1.37		6.12 ± 1.87	
Emirate of residence *	Abu Dhabi	3.46 ± 1.30	0.046	5.28 ± 2.42	0.090
	Dubai	3.43 ± 1.17		5.65 ± 2.41	
	Sharjah	3.33 ± 1.05		5.57 ± 2.42	
	Ajman	3.18 ± 1.19		5.74 ± 2.53	
	Umm Al Quwain	4.08 ± 1.12		5.12 ± 2.40	
	Ras Al Khaimah	3.58 ± 1.35		5.37 ± 1.95	
	Fujairah	3.50 ± 1.38		3.67 ± 2.61	
Residential area	Urban	3.41 ± 1.17	0.386	5.60 ± 2.38	0.584
	Rural	3.50 ± 1.27		5.48 ± 2.67	
Marital status	Single	3.38 ± 1.19	0.017	5.53 ± 2.46	0.082
	Married	3.65 ± 1.21		5.90 ± 2.23	
Degree	Undergraduate	3.42 ± 1.18	0.800	5.65 ± 2.46	0.002
	Postgraduate	3.45 ± 1.28		4.94 ± 2.10	
Employment	Employed	3.51 ± 1.21	0.289	5.11 ± 2.26	0.002
	Unemployed	3.40 ± 1.19		5.69 ± 2.46	

* Reported results are based on one-way ANOVA; others are based on a *t*-test.

**Table 4 nutrients-14-04175-t004:** Multiple regression models to identify the determinants of healthy habits.

	All Participants (n = 1003)		PNH Participants Excluded (n = 861)	
	Coefficient	*p*-Value	Coefficient	*p*-Value
Demographic Information				
Age	0.001	0.957	−0.007	0.623
Gender (Female)	−0.077	0.677	−0.101	0.586
Degree (Undergraduate)	0.020	0.877	0.025	0.858
Nationality (UAE National)	−0.121	0.559	0.107	0.632
Employment (Employed)	0.093	0.370	0.120	0.275
Marital status (Married)	0.115	0.063	0.110	0.101
Education and Nutritional Knowledge		<0.001 *		<0.001 *
Public Health and Nutrition major (PNH)	0.226	0.042	-	-
Received education related to healthy food and sustainable nutrition	0.233	0.006	0.077	<0.001
Nutritional knowledge	0.075	<0.001	0.232	0.008
Perceived barriers and contributing factors		0.078 *		0.170 *
Financial reasons	−0.013	0.888	−0.057	0.565
Time constraints	0.057	0.461	0.011	0.895
Busy rapid lifestyle	−0.040	0.604	−0.009	0.912
Inconvenience	0.111	0.246	0.106	0.296
I don’t know how to plan a diet.	−0.180	0.025	−0.138	0.106
I don’t know any easy recipes.	−0.106	0.228	−0.158	0.093
I’m not interested.	−0.188	0.076	−0.155	0.156
I don’t see the importance.	0.145	0.355	0.198	0.222

Dependent Variable: Healthy habits. * *p*-value calculated based on the partial F-test.

**Table 5 nutrients-14-04175-t005:** Multiple regression models to identify the determinants of nutritional knowledge.

	All Participants (n = 1003)		PHN Participants Excluded (n = 861)	
	Coefficient	*p*-Value	Coefficient	*p*-Value
Demographic information		<0.001 *		
Age	0.072	0.004	0.062	0.022
Gender (Female)	0.738	0.040	0.801	0.026
Degree (Undergraduate)	−0.512	0.045	0.535	0.046
Nationality (UAE National)	0.452	0.263	−0.425	0.331
Employment (Employed)	−0.433	0.032	−0.258	0.231
Marital status (Married)	0.156	0.194	0.146	0.263
Education		<0.001 *		
Nutrition major	1.422	<0.001	-	-
Received education related to healthy food and sustainable nutrition	0.577	0.005	0.462	0.028
Source of nutritional knowledge		<0.001 *		
School curriculums	0.516	0.004	0.444	0.024
Nutritionist/dietitian	0.048	0.789	0.121	0.544
I did my own research	0.731	<0.001	0.710	0.000
Social media	0.530	<0.001	0.497	0.004
I haven’t learnt about it	0.551	0.060	0.308	0.315

Dependent Variable: Nutritional Knowledge. * *p*-value calculated based on partial F-test.

**Table 6 nutrients-14-04175-t006:** Perceptions and opinions.

Categories		Count	Percentage %
Do you believe that healthy and sustainable nutrition is easily accessible to you?	Yes, very much	415	41.5%
	Somewhat	516	51.5%
	No, I don’t think it is	70	7.0%
Do you find information about healthy and sustainable nutrition easy to find and understand?	Yes, I do	753	75.2%
	No, I don’t	248	24.8%
What motivates you to eat healthy food?	Aesthetic reasons	322	32.1%
	To maintain a healthy weight	703	70.1%
	To keep my body healthy for a long time	728	72.6%
What do you think would help you and others eat healthier and more sustainable food?	Easy healthy recipes	317	31.6%
	Ready-to-eat healthy meals	527	52.5%
	Cheaper options	452	45.1%
	More variety	529	52.7%
	More availability	278	27.7%
	More knowledge and information	217	21.6%
	Personal one-on-one sessions	114	11.4%

## Data Availability

The data presented in this study are available on request from the corresponding author. The data are not publicly available due to privacy reasons.

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
