# Peer review of "The Barriers to Sustainable Nutrition for Sustainable Health among Zayed University Students in the UAE"

_nutrients, 2022, doi:10.3390/nu14194175_

Round 1

Reviewer 1 Report

Abstract: 2 last sentences of the abstract do not seem to be supported by this study.   Introduction: Few references in the introduction. Some statements are not supported by enough references, for example paragraph L50-54   Methods : -It is not clear on what basis the authors defined the number of points to be attributed to each nutritional component. It is not obvious that each component should have the same number of point. Moreover, some of the references for some of the components are from other countries. Is it relevant to use references from other countries ? Why not use the dietary guidelines of the country in which the study has been conducted ? Eating 1-2 fruit per day and eating 1-2 vegetable per day, is considered as a healthy habit. This seems to be a low threshold. -The statistical analysis section should be detailed : is it one regression model with all the variables or one model for each variable ? What are the adjustment factors ? -The article mentions several times "sustainable" nutrition, is it defined in the questionnaire ? What are the components of sustainable nutrition ? -More references could be added L106-107, this phenomena has been observed in other countries and a large body of literature is available (Freshman effect)   Results : Titles of tables should be revised to include basic information such as frequency (N=), place of the study, name of the study... It would be interesting to indicate the global p-value for each variable (e.g. p-value for the variable Education as a whole). Table 6 : Are all these variables included at the same time in the model or in separate model ? Is there a strategy for the selection of variables ?   Given the fact that knowledge seem to be an important factor, is it relevant to keep the students who study Nutrition in the analysis ? A separate analysis on the ones who don't study Nutrition could be interesting.   Discussion : Are there dietary guidelines/nutritional recommendations in UAE ? Is the population aware of these? This could be discussed.   Minor comments : L53 typo L290 missing word

Author Response

Dear Reviewer 

Kindly find attached our responses.

Many thanks and much appreciated

Regards

Sharifa 

Reviewer 2 Report

Manuscript ID: nutrients-1896284 - Review Report

This manuscript aims to identify barriers between knowledge and awareness of the importance of a healthy diet and lifestyle, and the practice of healthy and sustainable diets in a sample of Zayed university students in the United Arab Emirates.

The topic is interesting because it relates to diets sustainability and how to reach this goal, and younger generations are certainly the key population on which to target information and education interventions. Numerous surveys have been conducted worldwide on adolescents' and young adults' knowledge and awareness of food and environmental sustainability issues, and readiness for lifestyle change and new personal and community actions. Scarcer is research on perceived drivers and barriers to healthy and sustainable eating and how these relate to actual eating habits. This manuscript tries to fill this gap.

The manuscript is, all in all, well-conceived, however, it should be improved in the description of methods and presentation of results since both need to be more accurate and complete.

Below are my general comments for each paragraph.

Sampling. Authors should specify if a minimum number of responses (sample units) were planned, including the drop-out rate. More detail should be added on how the sample was recruited. For example, by specifying: who was in charge of disseminating the link to questionnaire? How were the e-mail addresses and social media contacts of students obtained?

Data collection tool. Questions on nutritional knowledge are a selection of those in GNKQ: the 11 selected questions should then be listed in a table or, alternatively, detailed in the text, also describing the type and meaning of scores adopted for answers. Similarly, the questions used to assess food habits should be first described in this section, either in text or in table form, perharps moving Table 3 in the Methods and clarifying that “yes” answer was assigned a score 1 and “no” answer was assigned a score equal to 0.

Statistical analysis. Authors should give some more information on the number of questionnaires filled-in in total, if all were considered valid, or if a threshold was fixed for validity check, for example based on the number of missing answers on key aspects investigated.

For statistical tests, please specify if variables were preliminarily checked for normality before applying parametric statistics. Moreover, the significance level fixed should be indicated, for example 5% (p< 0.05).

The regression models adopted should be better described by specifying the dependent variables considered each time and the corresponding explanatory variables; calculation of (adjusted) R-square should be reported. Reference to the linear regression model should be reflected in tables’ titles or footnotes.

References. 18 references seem a little few. Some more reference con be find reading Scalvedi ML, Gennaro L, Saba A and Rossi L (2021) Relationship Between Nutrition Knowledge and Dietary Intake: An Assessment Among a Sample of Italian Adults. Front. Nutr. 8:714493.doi: 10.3389/fnut.2021.714493

Below my comments for specific lines.

Line 14 and Line 44. UAE should be written out in full the first time it appears in the text, putting UAE in brackets, and use acronym in the subsequent texts.

Lines 34-35. Please, rephrase this sentence, I would avoid repeating “goals” here, for example “a diet that is nutritionally sufficient based on safe and healthy food”.

Line 45. “Data” instead of “Records”.

Line 68. “pre-made” should be replaced by “pre-cooked” or “ready-to-eat” or “conventional” food.

Line 83. Instead of “that are being done” use “currently underway”.

Line 136-137. Please rewrite the sentence, for example “...were carried out to compare mean scores on nutritional habits and nutritional knowledge across subgroups of sociodemographic variables”.

Line 156-157. “Art-related majors made up combined” should be “Art-related majors made up a combined group”?

Line 168. Table 2 should be eliminated; data are already reported in text.

Line 173-179. All this period pertains more to Methods than to Results.

Line 181. Table 3 with questions on eating habits and scoring should be moved in Methods, as already commented above.

Line 182. Table 4 should be eliminated; its content is already written in text (line 173).

Line 195 and 223. The paragraphs have the same title, please correct.

Author Response

Dear Reviewer,

Kindly find attached our responses.

Many thanks and much appreciated.

Regards

Sharifa  
